# Polarization Splitting at Visible Wavelengths with the Rutile TiO_2_ Ridge Waveguide

**DOI:** 10.3390/nano13121891

**Published:** 2023-06-20

**Authors:** Xinzhi Zheng, Yujie Ma, Chenxi Zhao, Bingxi Xiang, Mingyang Yu, Yanmeng Dai, Fang Xu, Jinman Lv, Fei Lu, Cangtao Zhou, Shuangchen Ruan

**Affiliations:** 1Shenzhen Key Laboratory of Ultraintense Laser and Advanced Material Technology, Center for Advanced Material Diagnostic Technology, College of Engineering Physics, Shenzhen Technology University, Shenzhen 518118, China; 2110416034@stumail.sztu.edu.cn (X.Z.); 2200413002@stumail.sztu.edu.cn (C.Z.); yumingyang@sztu.edu.cn (M.Y.); daiyanmeng@sztu.edu.cn (Y.D.); xufang@sztu.edu.cn (F.X.); lvjinman@sztu.edu.cn (J.L.); zcangtao@sztu.edu.cn (C.Z.); 2College of Application and Technology, Shenzhen University, Shenzhen 518118, China; 3College of New Materials and New Energies, Shenzhen Technology University, Shenzhen 518118, China; xiangbingxi@sztu.edu.cn; 4School of Information Science and Engineering, Shandong University, Jinan 250100, China; lufei@sdu.edu.cn

**Keywords:** on-chip polarization splitting, r-TiO_2_ thin film, optical ridge waveguide, micro-ring resonator

## Abstract

On-chip polarization control is in high demand for novel integrated photonic applications such as polarization division multiplexing and quantum communications. However, due to the sensitive scaling of the device dimension with wavelength and the visible-light absorption properties, traditional passive silicon photonic devices with asymmetric waveguide structures cannot achieve polarization control at visible wavelengths. In this paper, a new polarization-splitting mechanism based on energy distributions of the fundamental polarized modes in the r-TiO_2_ ridge waveguide is investigated. The bending loss for different bending radii and the optical coupling properties of the fundamental modes in different r-TiO_2_ ridge waveguide configurations are analyzed. In particular, a polarization splitter with a high extinction ratio operating at visible wavelengths based on directional couplers (DCs) in the r-TiO_2_ ridge waveguide is proposed. Polarization-selective filters based on micro-ring resonators (MRRs) with resonances of only TE or TM polarizations are designed and operated. Our results show that polarization-splitters for visible wavelengths with a high extinction ratio in DC or MRR configurations can be achieved with a simple r-TiO_2_ ridge waveguide structure.

## 1. Introduction

On-chip polarization handling devices play important roles in many applications, including polarization diversity/(de)multiplexing for realizing polarization-transparent photonic integrated circuits (PICs) [1,2,3], polarization division multiplexing (PDM) [4,5], coherent optical communications [6,7,8,9], and quantum photonics integrated circuits [10]. Silicon nanophotonic waveguides with ultrahigh-index contrast can achieve birefringence as high as 0.7 of two fundamental modes [11,12], making possible fabrication of ultracompact on-chip polarization handling devices (OCPHDs) [9]. Novel silicon-on-insulator (SOI) OCPHDs, but only for the infrared (IR) regime, have been proposed [9,10,11,12]. However, modern quantum PIC applications often require polarization manipulation at visible wavelengths [13,14]. Integrated photonic platforms operating at visible wavelengths are of great interest for applications ranging from quantum optics and metrology to bio-sensing and biomedicine [15,16,17]. Furthermore, visible wavelength light is used for quantum state preparation [18], manipulation [19], readout of color centers [20], quantum dots [21], and various quantum emitters in 2D materials [22,23]. These applications will inevitably involve polarization-sensitive devices, so it is necessary to develop on-chip PBSs for visible wavelengths. For example, visible light polarization control on-chip can be applied in quantum systems and underwater datacom for advanced cooling schemes for quantum computing [24].

The polarization beam splitter (PBS) is regularly used for separating/combining TE and TM mode polarizations. Most existing PBSs work in the near-infrared regime and require rather demanding fabrication. Various structures have been proposed to realize on-chip PBSs [11,25,26,27,28,29,30,31,32,33,34,35,36,37,38,39,40,41], such as those in multimode interferometers [25,26], the Mach–Zehnder interferometer (MZI) [27,28,29,30,31], photonic crystals (PhC) [32,33,34,35], mode evolu tions [36], arrayed-waveguide gratings (AWGs) [37], and the plasmonic waveguide [38]. Compared with such structures, directional couplers (DCs) and asymmetric direct couplers (ADCs) for PBSs are more convenient due to their structural and design simplicity [11,39,40,41]. The performance of DCs and ADCs relies on the distinct birefringence of two fundamental modes in silicon waveguides resulting from the geometrical asymmetry in the sub-micrometer cross-section. However, because of the visible-light absorbing properties of silicon, as well as the dimension-scaling requirements of silicon nanowires, SOI PBS operating at visible wavelengths is still unavailable. Transparent materials, such as lithium niobite (LN) and titanium dioxide (TiO_2_), have relatively low refractive index compared with silicon, but it is difficult to separate the fundamental modes’ polarizations using birefringence by adjusting the geometrical asymmetry for the visible wavelengths. Accordingly, a polarization-control device with a simple waveguide structure for visible wavelengths is desirable.

Titanium dioxide (TiO_2_) is promising for realizing visible-spectrum integrated photonic devices due to its favorable optical properties such as relatively high refractive index, large bandgap, low thermal expansion coefficient, negative thermo-optic coefficient [42,43,44], and large Kerr nonlinearity [45,46]. Moreover, bio-compatibility and environmental friendliness allow its wide use [47]. Among its three (anatase, rutile, and brookite) phases, TiO_2_ in the rutile phase (r-TiO_2_) has the largest (>2.7) refractive index as well as the best thermal stability. Recently, we successfully fabricated a heterostructure composed of an r-TiO_2_ thin film on a SiO_2_ substrate through ion implantation combined with Cu-Sn bonding [48], thereby opening a door for achieving on-chip polarization-handling devices for the visible spectrum. However, in order to realize optical waveguides on r-TiO_2_ thin films using micromachining techniques, it is first necessary to estimate the workable and optimum structural parameters of the waveguide.

We propose here a novel polarization-splitting mechanism based on a simple ridge waveguide structure that is applicable to visible light. The characteristics of bending loss with different bending radii and optical coupling properties for the fundamental polarization modes in certain selected r-TiO_2_ ridge waveguide configurations, as well as the polarization-splitting mechanism involving different energy distributions of the TE0 and TM0 modes in the ridge waveguide, are analyzed. A polarization-splitter with a high extinction ratio operating for visible wavelengths based on DCs in the r-TiO_2_ ridge waveguide structure, with a transmission extinction ratio of −40 dB for the TE/TM polarization beam splitter, is used. We also propose an MRR-based polarization-selective filter in the r-TiO_2_ ridge waveguide structure. For the TM MRR, the transmission of the TE0 polarization mode at the drop port is reduced to −58 dB. For the TE MRR, the transmission of the TM polarization mode at the drop port is reduced to −40 dB. Our results suggest that a polarization splitter with a high extinction ratio for the visible wavelengths in DC or MRR configurations can be achieved by tailoring the structural dimension of the r-TiO_2_ ridge waveguide.

## 2. Materials and Methods

The ridge waveguide is widely used in various integrated optical devices [49]. As shown in Figure 1a, it consists of a slab with a superimposed strip, where H and h are the rib and slab heights, respectively, and W is the rib width. According to the simulation results for optical transmission in the r-TiO_2_ ridge waveguide based on the 3D-FDTD model, we found that the optical energy distribution of the two fundamental modes (TE0 and TM0) in the r-TiO_2_ ridge waveguide structure is distinct. As shown in Figure 1a,b, more TE0 energy tends to propagate in the lower slab layer, and most TM0 energy tends to be in the rib strip. Due to these energy distribution properties, if the waveguide is bent, the TE0 mode tends to leak from the lower layer, and it is easily coupled to the waveguides adjacent to the latter, as shown in Figure 1c,d. The difference in the energy distributions of the TE0 and TM0 modes can thus be used for the polarization separation of light. Moreover, this polarization-splitting mechanism is applicable to the visible spectrum, suggesting its use in on-chip polarization-handling devices for visible wavelengths.

In order to effectively separate the TE0 and TM0 modes for optimizing polarization-splitting performance, the width and thickness of the r-TiO_2_ ridge waveguide structure should be tailored. To distinguish the two fundamental modes, their distribution difference should be enhanced. Accordingly, h/H should be as small as possible in order to inhibit TM0 mode propagation in the lower slab, and W/H should be as small as possible in order to inhibit TE0 mode propagation in the top rib strip. Moreover, to prevent interference by the higher-order modes, the ridge waveguide should be restricted to single-mode propagation, i.e., h/H should not be too small. A too-small h/H would lead to the waveguide height increasing, so higher-order mode interactions can occur. Meanwhile, a too-small W/H would make it difficult for light to propagate in the waveguide. Therefore, there should be a threshold value for both h/H and W/H to guarantee single-mode propagation in the r-TiO_2_ ridge waveguide, and the r-TiO_2_ ridge waveguide structural parameters should be appropriately tailored. In order to maintain low-loss, single-mode transmission and retain the TE0 mode in the lower slab layer as much as possible, we found that for W = 0.25 μm rib width and 0.6 μm r-TiO_2_ film thickness, the minimum value of h should be 0.16 μm.

The mode energy distribution of the r-TiO_2_ ridge waveguide, obtained using the finite difference eigenmode (FDE) code [50], which is based on solving Maxwell’s equations for a cross-sectional mesh in the waveguide, is shown in Figure 1a,b, where the effective refractive indices used are *n*(TiO_2_) = 2.72 (core), *n*(SiO_2_) = 1.444 (substrate), and *n*(air) = 1 (cladding). To calculate the bending loss of polarized light in the curved r-TiO_2_ waveguide, the model conducted using FDTD solutions from Lumerical software and the 3D-FDTD algorithm were used for simulation, and the detailed calculation mechanism is shown in Refs. [51,52]. The light source in Figure 1c,d is the fundamental TE0 mode and fundamental TM0 mode, respectively. The mesh accuracy is 20 nm, and all boundary conditions are set to be PML conditions. There are mainly three types of propagation loss in the r-TiO_2_ waveguide, which are scattering loss, absorption loss, and bending loss. The absorption loss can be negligible in the simulation model, and the bending loss that depends on the waveguide structure, especially the radius of the curved waveguide, is discussed in the following section. The scattering loss attributed to the scattering of propagating light from the rough side wall of the waveguide can be simulated by setting the meshing, and the scattering loss occurs when the simulation meshing is 20 nm. According to the experimental results in Ref. [53], the propagation loss in the strip waveguide with a width of 250 nm and a height of 70 nm on the amorphous TiO_2_ thin film was determined to be 7.5 dB/cm. In comparison, the propagation loss calculated to be around 1.1 dB/cm in the r-TiO_2_ ridge waveguide proposed in this paper is reasonable. Electron beam lithography (EBL) and inductively coupled plasma etching (ICP) were used to fabricate the ridge waveguide structure on the r-TiO_2_ thin film, and the fabrication process is briefly demonstrated here. Firstly, chromium and photoresist thin films are deposited on the sample surface successively, and the waveguide pattern is exposed and developed on the photoresist. Then, the pattern of the photoresist is transferred to the r-TiO_2_ thin film using ICP etching technology, and finally, the etching effect is observed using a scanning electron microscope (SEM).

As shown in Figure 2a,b, the bending loss in the TE0 and TM0 modes decreases with an increasing radius of the curved waveguide. In the curved r-TiO_2_ ridge waveguide, the bending loss in the TE0 mode is clearly larger than that in the TM0 mode, especially when the radius is above 4 μm. The bending loss in the TM0 mode decreases dramatically as the bending radius increases from 2 μm to 4 μm, but gradually decreases with a further radius increase. When the radius is 4 μm, the bending loss in the TM0 mode is around 0.007 dB/μm, but that in the TE0 mode is about 1.43 dB/μm. Thus, the TM0 mode can propagate in this curved r-TiO_2_ waveguide with a radius of 4 μm with very low bending loss, but very little TE0 mode energy is retained due to its large bending loss when transmitting through the bending structure. That is, the two fundamental modes can be well separated using a bending r-TiO_2_ waveguide with an appropriate radius. Furthermore, Figure 2c,d shows that varying the rib width W and slab height h has a negligible effect on the bending loss. In particular, the bending loss remains almost the same for the TM0 mode, and that in the TE0 mode is also limited. In fact, the size of a bending waveguide can be increased without affecting the bending loss. Accordingly, a curved r-TiO_2_ waveguide structure, such as the MRR, can be designed to separate the TE0 and TM0 modes since the TE0 mode tends to leak from the lower slab layer.

The difference in the coupling efficiencies of the TE0 and TM0 modes in r-TiO_2_ ridge waveguides was also investigated. The coupling coefficient was obtained using the Lumerical 3D-FDTD simulation code, which is based on the 3-dimensional solution of the Maxwell equations. Figure 3a,b shows that the TE0 and TM0 modes in the two adjacent r-TiO_2_ waveguides are coupled, but they are out of phase. The energy in the TE0 mode is of the period L_TE_ = 15 μm, and that in the TM0 mode is L_TM_ = 60 μm. Therefore, the coupling times of the two fundamental modes in the two adjacent waveguides can be modulated by changing the coupling length(s) of the waveguides. For example, Figure 3a,b shows that if the coupling length is 60 μm and the coupling distance is 0.2 μm, the TM0 mode will be coupled from waveguide 1 to waveguide 2, and the TE0 mode will be coupled from waveguide 1 to waveguide 2 twice: eventually back to waveguide 1, i.e., “over-coupling” occurs. Therefore, the two fundamental modes can be separated into different waveguides by tailoring the coupling parameters. Figure 3c,d shows the corresponding transmission spectra of the TE0 and TM0 modes in the two adjacent r-TiO_2_ waveguides, respectively. Moreover, the coupling periods of the TE0 and TM0 modes increase with the coupling distance. From these results, we can see that the coupling length and distance in the two adjacent r-TiO_2_ waveguides can be controlled in order to selectively separate the two fundamental modes.

## 3. Results and Discussion

The PBS is one of the most important polarization-handling devices for separating/combining TE and TM polarizations. Here, a PBS based on the DC mechanism that can be operated in the visible spectrum is proposed. This PBS, consisting of a TM polarization beam splitter and a TE polarization beam splitter in the r-TiO_2_ ridge waveguide, is designed according to the analysis above and shown in Figure 4. Light enters the main (straight) waveguide through the input port, and the TM0 and TE0 modes are separated and coupled to the polarization beam splitters. The TM polarization beam splitter consists of a curved waveguide connected to two straight waveguides, and its width is W1 = 0.25 μm. According to the above analysis, in order to separate two fundamental modes as much as possible, the coupling length L1 and the coupling gap1 between the TM polarization beam splitter and the main straight waveguide are 60 μm and 0.2 μm, respectively. Most of the TM0 mode will then be coupled to the TM polarization beam splitter. The TE0 mode is excluded from the latter due to over-coupling, as discussed. Instead, it propagates along the main straight waveguide and is eventually coupled to the TE polarization beam splitter. In the TM polarization beam splitter, a curved waveguide structure with bending radius R1 = 4 μm is used to further remove the TE0 mode, which suffers high bending loss (1.43 dB/μm). The bending loss in the TM0 mode in the curved waveguide is only 0.007 dB/μm, so that the TM0 mode can propagate through the curved waveguide with little loss and exit from the Cross port. For the TE polarization beam splitter, two parallel coupled straight waveguides with different lengths are used to couple to the main waveguide. The coupling gap2 is 0.3 μm, and the coupling length L2 is =30 μm. With these coupling parameters, the TE0 mode propagating along the main waveguide will be coupled to waveguide 1, and then to waveguide 2, and be detected at the Through port. This TE polarization beam splitter with three parallel straight waveguides can effectively inhibit the TM0 mode from being coupled to the straight waveguide 2 because of its weak coupling efficiency.

The transmission of the TE0 mode and TM0 mode energy in the proposed r-TiO_2_ PBS were numerically simulated, and the results are shown in Figure 5a,b. In the TM polarization beam splitter shown in Figure 5a, the TM0 mode is coupled from the main straight waveguide to the TM polarization beam splitter and propagates along the waveguide with low loss. In contrast, the TE0 mode is coupled between the main and adjacent straight waveguides of the TM polarization beam splitter four times and then back to the main straight waveguide. It is then coupled to the TE polarization beam splitter, and output from the Through port. The transmitted spectrum at the Cross and Through ports are shown in Figure 5c,d, respectively. We see that for 610 to 660 nm light, the largest polarization extinction ratio for both the TM and TE beam splitters in our PBS reaches almost −40 dB, and polarization splitting of the TE beam splitter is more stable than that of the TM beam splitter. The polarization extinction ratio of the TE0 and TM0 modes in the TM polarization beam splitter can be enhanced by increasing the length of the curved waveguide. Thus, polarization separation with a high extinction ratio for the visible spectrum can be achieved with the r-TiO_2_ ridge waveguide PBS.

The micro-ring resonator (MRR) is a basic component of optical platforms [54,55]. Because of its ultrahigh quality factor and small mode volume, MRR is useful in integrated and optical devices as well as information processing technologies, including wavelength division multiplexing (WDM) and polarization division multiplexing (PDM). Here we present a polarization-selective filter based on an add–drop type MRR in the r-TiO_2_ ridge waveguide structure. The add–drop type MRR consists of a micro-ring waveguide coupled to two access waveguides with four ports, namely the Input, Through, Add, and Drop ports. For an MRR with bending radius R, the circumference of the micro-ring waveguide is l=2πR, and the resonance wavelength is $\lambda =n_{eff}l/m$. Assuming that the coupling regions on both sides of the micro-ring are the same, i.e., $t_{1}=t_{2}=t$ and $k_{1}=k_{2}=k,$ where $k_{1}$ ($k_{2}$) is the coupling coefficient and $t_{1}$ ($t_{2}$) the transmittance coefficient in the coupling region consisting of the input and the micro-ring waveguides (the output and the micro-ring waveguides), the peak power $T_{drop}$ of the transmission is
(1)$$T_{drop}=\frac{k^{4}\alpha }{{\left(1-\alpha t^{2}\right)}^{2}}$$
so that the peak power $T_{drop}$ approaches zero for
(2)$$k^{2}\alpha^{1/2}\ll 1$$

Therefore, in order to achieve MRR-based, polarization-selective optical filters, one can manipulate the polarization-dependent bending loss and coupling ratio of the MRRs by tailoring the waveguide structure. Resonance in the undesired polarization mode in the ring can be suppressed by minimizing its evanescent coupling and/or enhancing its bending loss. For the desired polarization mode, the coupling ratio and the bending loss should be optimized to satisfy the low loss and high extinction ratio. That is, a polarization beam splitter with an ultra-high extinction ratio can be achieved by controlling the structural parameters of the r-TiO_2_ MRR. Based on the above numerical analysis, an r-TiO_2_ MRR-based, TM-polarization-selective optical filter is proposed. An r-TiO_2_ MRR in a ridge waveguide with radius R = 4 μm can result in a bending loss of 0.007 dB/μm for the TM0 mode and 1.43 dB/μm for the TE0 mode for 635 nm light transmission around the ring. That is, the TM0 mode can propagate with low bending loss, and the TE0 mode decays, in the micro-ring waveguide. In order to achieve an ultra-high polarization extinction ratio, the coupling coefficient of the TE0 and TM0 modes between the straight and the ring waveguides should be precisely adjusted. Accordingly, the coupling strengths of the TE0 and TM0 modes for different coupling lengths were calculated, and the results are shown in Figure 6c. Based on these simulation results, a racetrack resonator, as shown in Figure 6a, was designed. The coupling length is 40 μm and the coupling distance gap1 is 0.25 μm. The coupling coefficients $k_{TE}$ and $k_{TM}$ are 0.04 and 0.35, respectively. For this MRR-based TM0 polarization filter, we have $k^{2}\alpha^{1/2}=3\times 10^{-5}\ll 1.0$ for the TE mode, thus satisfying Equation (2). That is, the TE0 is inhibited from propagating in this racetrack resonator, but the TM0 mode can be resonant in the MRR and exit from the Drop end as output.

The spectral response of our polarization splitter was obtained using the transmission matrix method [56] and the three-dimensional finite difference time domain method. Figure 7a,b shows the transmission spectra at the Through and Drop ends when only TM and TE polarization light, respectively, of a wavelength ranging from 610 nm to 660 nm enters the waveguide. We see that there is ultra-low crosstalk between the TE0 and TM0 modes. Thus, according to the simulations, this MRR-based, TM-polarization-selective optical filter can realize a low loss and low crosstalk ground filtering for TM0 input light. In contrast, the polarization extinction ratio of the TE0 polarization mode can reach −58 dB, and resonance is suppressed, but the input TE0 mode can easily propagate through the straight waveguide and exit from the Through end with low loss.

Due to the coupling readiness of the TE0 mode compared with the TM0 mode, the two fundamental polarized modes can be selectively separated by adjusting the structural parameters of the two adjacent ridge waveguides. Accordingly, we propose an r-TiO_2_ MRR-based, TE-polarization-selective optical filter, as shown in Figure 8, whose coupling length and coupling distance can be tailored to inhibit coupling of the TM0 mode and ensure complete coupling of the TE0 mode from the straight waveguide to the racetrack waveguide.

According to the simulation results shown in Figure 8c, if the coupling length is 3 μm and the coupling distance gap1 is 0.35 μm, the coupling coefficients are $k_{TE}=0.13$ and $k_{TM}=0.008$. For this MRR-based, TE-polarization-selective optical filter, the TM polarization mode has $k^{2}\alpha^{1/2}=6.4\times 10^{-5}$, which is much less than 1.0 and thus satisfies Equation (2). This means that the TM0 mode is effectively inhibited from resonance in this polarization splitter, passing through the straight waveguide and output from the Through end with low loss. In order to reduce the bending loss in TE0 mode in the racetrack waveguide, the dimension width W2 of the ring ridge waveguide was increased to 0.5 μm and the radius R1 was 10 μm, in which the bending radiation loss in TE0 is less than 2 dB/cm and the cross-talking of higher modes could be negligible. To solve the width mismatch in the straight waveguide and bent waveguide in the racetrack waveguide, a tapered waveguide with a length L1 of 5 μm was added between the straight waveguide and bent waveguide, as shown in Figure 8. This designed racetrack resonator can reduce the size of the ring resonator and increase the FSR.

The spectral response of our polarization splitter is shown in Figure 9a,b for the transmission spectra of 610 nm to 660 nm light at the Through and Drop ends when only TE and TM polarization light, respectively, are input into the waveguide. The TE polarization filter has a typical micro-ring resonance effect with low loss and low crosstalk when the input light is TE polarized. In contrast, when the input light is TM polarized, the transmittance peaks at the Drop end are as low as −40 dB, indicating TM mode resonance is effectively suppressed. With this MRR design, the TM mode can pass through the straight waveguide and exit from the Through end with low loss.

A brief summary of previously reported PBSs based on different structures is presented in Table 1. It can be seen that the proposed PBS in this work is working at visible wavelengths, which is different from previous PBSs.

## 4. Conclusions

In this work, a novel polarization-splitting mechanism based on the different energy distributions of the fundamental polarization modes in r-TiO_2_ ridge waveguides is proposed. The characteristics of bending loss for different bending radii and the optical coupling properties for the fundamental modes in r-TiO_2_ ridge waveguide configurations are analyzed. A polarization splitter, based on DCs in the r-TiO_2_ ridge waveguide structure, with a high extinction ratio for visible wavelengths, is accordingly designed. The TE0 mode transmission at the Cross port in the TM polarization beam splitter is suppressed to as low as −40 dB, and the polarization extinction ratio of the TM0 mode at the Through port in the TE polarization beam splitter could exceed −40 dB. In addition, polarization-selective filters based on MRRs that operate only for the resonance of TE or TM polarizations are designed. For the TM-type MRR, the transmission of the TE0 mode at the Drop port is suppressed to only −58 dB. For the TE-type MRR, the transmission of the TM0 mode at the Drop port is suppressed to only −40 dB. Our results suggest that a polarization splitter with a high extinction ratio working at both visible and infrared regimes in DC or MRR configurations can be achieved with simple r-TiO_2_ ridge waveguide structures.

## Figures and Tables

**Figure 1 nanomaterials-13-01891-f001:**
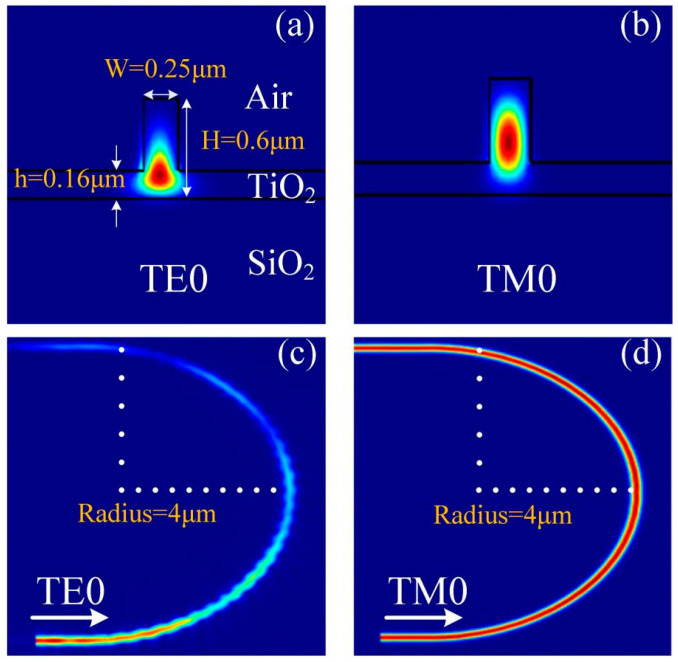
Energy distribution (arbitrary units; red: strongest, blue: weakest) in a ridge waveguide for the TE0 (**a**) and TM0 (**b**) modes at 635 nm. Energy distribution of the fundamental polarized modes propagating in a bent rib waveguide with the cross-section shown in (**a**) for the TE0 (**c**) and TM0 (**d**) modes, respectively. The radius of the bent waveguide is 4 μm.

**Figure 2 nanomaterials-13-01891-f002:**
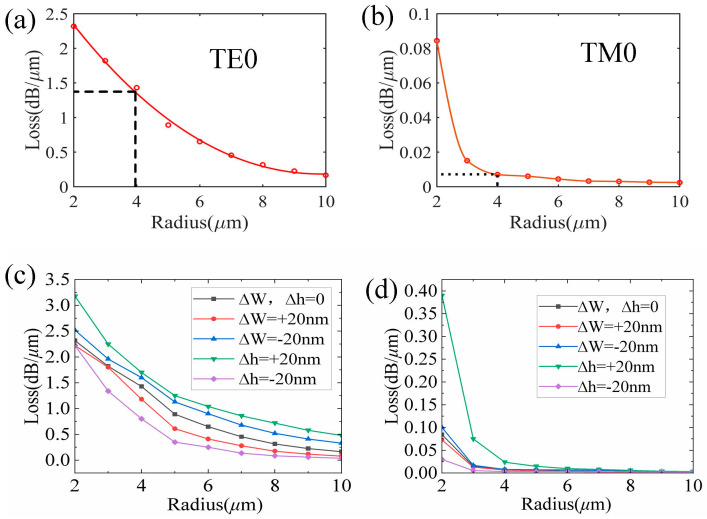
The bending loss in a 635 nm curved ridge waveguide for the TE0 mode (**a**) and TM0 mode (**b**) with different radii, and that for the TE0 mode (**c**) and TM0 mode (**d**) versus the waveguide width ∆W and slab height ∆h.

**Figure 3 nanomaterials-13-01891-f003:**
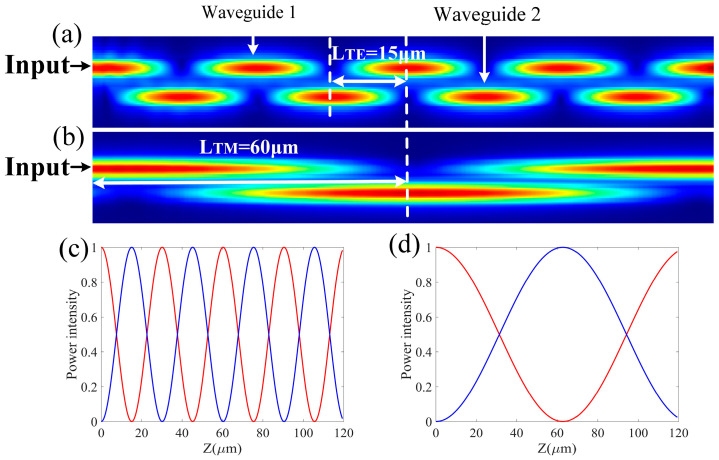
The electric fields of the TE0 (**a**) and TM0 (**b**) modes along the coupling length in the upper and lower waveguides, and the corresponding field intensities of the (**c**) TE0 and (**d**) TM0 modes in the upper (red curve) and lower (blue curve) waveguides. One can see that for both modes, the fields in the two adjacent waveguides are exactly out of phase, and there is no wave attenuation.

**Figure 4 nanomaterials-13-01891-f004:**
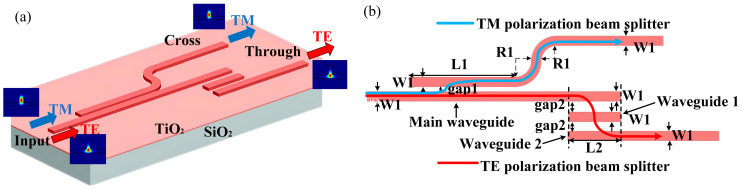
Schematic showing the proposed PBS (**a**) and its top view (**b**).

**Figure 5 nanomaterials-13-01891-f005:**
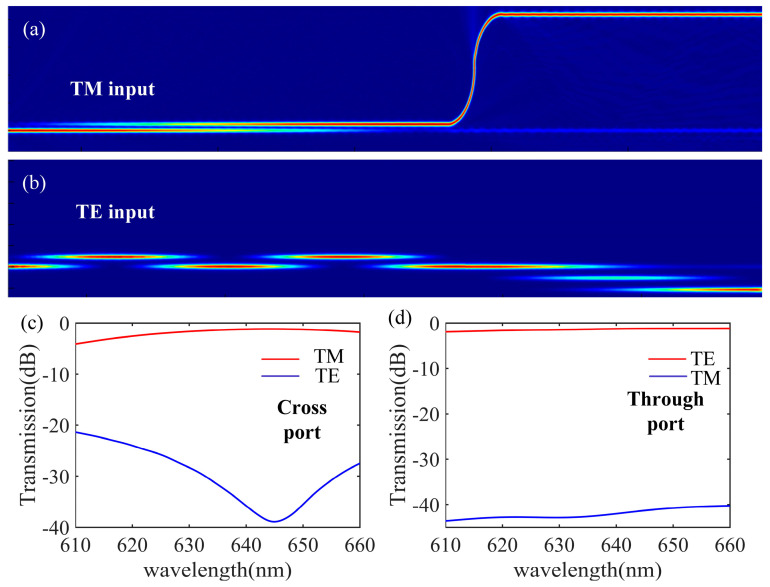
Optical field distribution of the r-TiO_2_ ridge waveguide PBS for the TM (**a**) and TE (**b**) modes. Calculated optical transmission of the Cross (**c**) and Through (**d**) ports for the TM and TE polarizations.

**Figure 6 nanomaterials-13-01891-f006:**
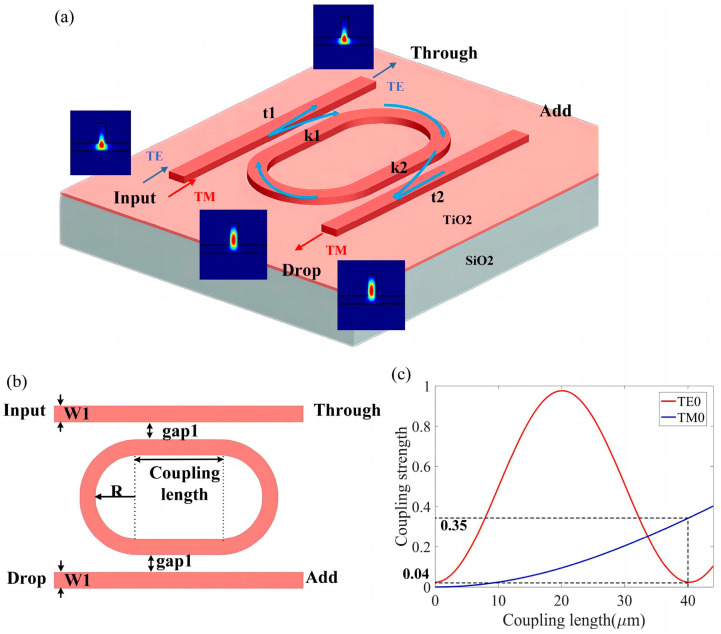
Schematic showing the proposed MRR-based TM polarization optical filter: (**a**) 3D view and (**b**) top view. (**c**) Coupling strength versus coupling length of the TE0 and TM0 modes, as calculated.

**Figure 7 nanomaterials-13-01891-f007:**
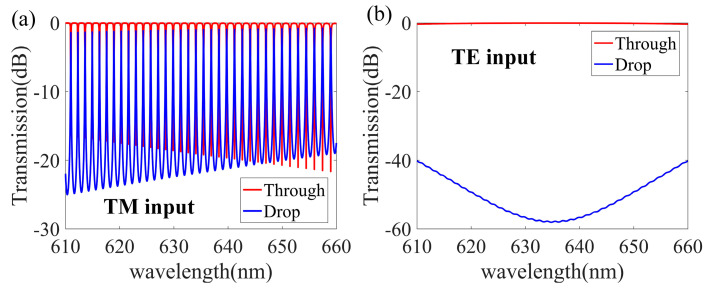
Simulated spectral responses for the designed TM-type polarization splitter at the Drop and Through ends when the input light is only TM polarization (**a**) and TE polarization (**b**), respectively.

**Figure 8 nanomaterials-13-01891-f008:**
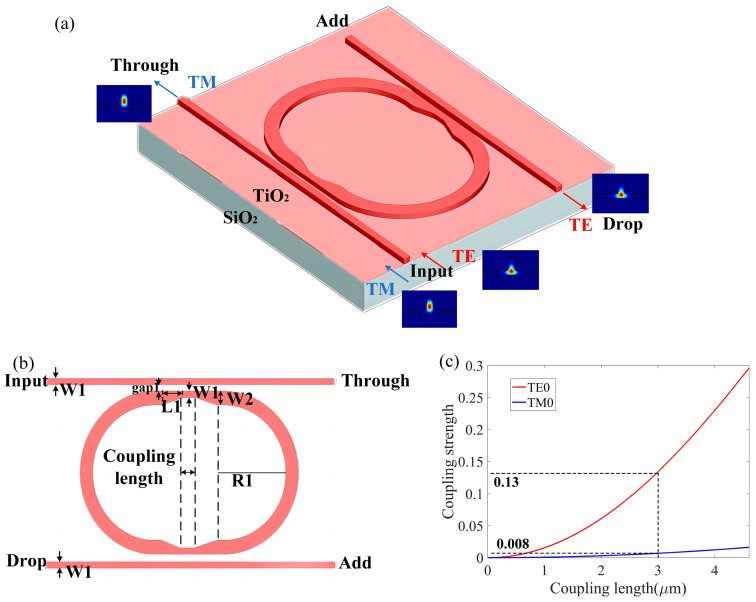
Schematic configuration for the MRR-based, TE-polarization-selective optical filter in a 3D view (**a**) and top view (**b**), and the calculated coupling strength of the TE0 and TM0 modes with different coupling lengths (**c**).

**Figure 9 nanomaterials-13-01891-f009:**
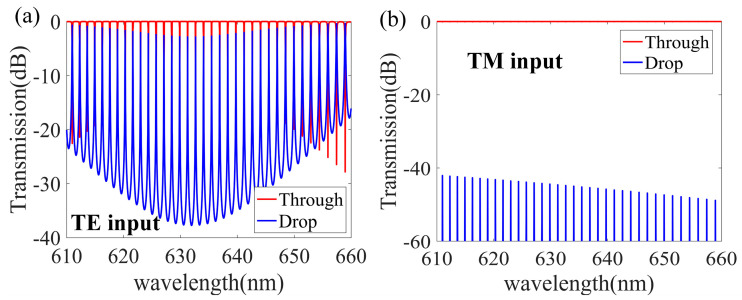
Simulated spectral responses in the designed TE-type polarization splitter at the Drop and Through ends when the input light is only TE polarization (**a**) and TM polarization (**b**), respectively.

**Table 1 nanomaterials-13-01891-t001:** Comparison of PBSs based on different structures.

Structure	Length (μm)	ER (dB)	EL (dB)	Bandwidth (nm)
LSPs [57]	1.1	20.69 (TE) 20.33 (TM)	0.249 (TE) 0.671 (TM)	(1450 nm–1650 nm) 200 nm (ER > 20 dB, TE) (ER > 12 dB, TM)
DCs [58]	11	20.32 (TE) 15.4 (TM)	1.69 (TE) 2.72 (TM)	(1530 nm–1565 nm) 35 nm (ER > 10.6 dB, TE) (ER > 14.6 dB, TM)
ADCs [59]	16	26.7 (TE) 21.3 (TM)	0.5 (TE) 0.2 (TM)	(1480 nm–1620 nm) 140 nm (ER > 10 dB, TE) (ER > 15 dB, TM)
SWGs [60]	160	25 (TE) 31 (TM)	<1	(1450 nm–1634 nm) 185 nm (ER > 20 dB, TE) (1490 nm–1575 nm)85 nm (ER > 20 dB, TM)
ADCs [61]	31	55 (TE) 55 (TM)	0.066 (TE) 0.2 (TM)	(1500 nm–1600 nm) 100 nm (ER > 29.4 dB, TE) (1527 nm–1587 nm)60 nm (ER > 20 dB, TM)
DCs (this work)	90	38 (TE) 43 (TM)	2 (TE) 0.45 (TM)	(610 nm–660 nm) 50 nm (ER > 40 dB, TE) (ER > 22 dB, TM)
MRRs (this work)	48	45 (TE) 58 (TM)	2.5 (TE) 0.7 (TM)	(610 nm–660 nm) 50 nm (ER > 40 dB, TE) (ER > 40 dB, TM)

## Data Availability

Not applicable.

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
