# Peer review of "Polarization Splitting at Visible Wavelengths with the Rutile TiO2 Ridge Waveguide"

_nanomaterials, 2023, doi:10.3390/nano13121891_

Round 1

Reviewer 1 Report

In this paper, the authors consider rutile-core ridge waveguides and propose several TE/TM mode-separating devices with simple structures that exhibit high extinction ratios for visible light, which has an impact in the field of integrated photonics. The data are based on detailed numerical calculations and have scientific validity. Therefore, I judge that this paper should be accepted after appropriate responses to my following remarks.

1) Although rutile thin films are transparent to visible light, there is no guarantee that the propagation loss of a waveguide with such a material as its core is zero. If you have any experimental findings or references on this, please add them.

2) In lines 114-115, the author says "the ridge waveguide should be restricted to single-mode propagation, i.e., h/H and W/H should not be too small." Is this not a sufficient explanation? Normally, to suppress higher-order modes, the core size should be reduced. However, too small a core size may cause cutoff.

Author Response

Dear Reviewer:

Thank you very much for your letter referring to the manuscript of "Polarization splitting at visible wavelengths with r-TiO2 ridge waveguide" (ID: 2436654). We have checked and modified our manuscript according to the comments from reviewers. The detail of modification is listed as follows:

For the comments proposed by reviewer 1:

1.The first review point:

“Although rutile thin films are transparent to visible light, there is no guarantee that the propagation loss of a waveguide with such a material as its core is zero. If you have any experimental findings or references on this, please add them.”

Response:

The propagation loss of an optical waveguide mainly includes scattering loss, absorption loss and bending loss. The scattering loss is attributed to the scattering of the propagating light from the rough side wall of the waveguide. Absorption loss is due to the optical absorption of the physical material. Bending loss is caused by the optical energy leakage when propagating in the curved waveguide light. As a matter of fact, the propagation loss has been taken into account in the numerical analysis process. The scattering loss caused by the rough waveguide sidewall is simulated through setting the meshing, and the scattering loss occurs when the simulation meshing is 20nm. The absorption loss is normally neglected in the simulation model. The bending loss that depends on the waveguide structure, especially the radius of the curved waveguide, was also analyzed in this paper. Therefore, the propagation loss of the r-TiO2 waveguide is not zero in the simulated optical transmission results in this paper, and we will make optical ridge waveguides on the fabricated r-TiO2 thin film to test the propagation loss of r-TiO2 waveguides in our future research work. In this paper, we have compared our simulated propagation loss in the r-TiO2 waveguides with some reported experimental results, in order to verify the accuracy of our results.

According to this suggestion, we add statements “There are mainly three types of propagation loss in the r-TiO2 waveguide, which are scattering loss, absorption loss and bending loss. The absorption loss can be negligible in the simulation model, and the bending loss that depends on the waveguide structure, especially the radius of the curved waveguide is discussed in the following part. The scattering loss attributed to the scattering of the propagating light from the rough side wall of the waveguide can be simulated through setting the meshing, and the scattering loss occurs when the simulation meshing is 20nm. According to the experimental results in Ref. [54], the propagation loss in the strip waveguide with the width of 250nm and the height of 70nm on the amorphous TiO2 thin film was tested to be 7.5dB/cm. In comparison, the propagation loss calculated to be around 1.1dB/cm in the r-TiO2 ridge waveguide proposed in this paper is reasonable.” at the lines from 140 to 150 in the third paragraph of Part 2, marked in red.

  1. The second review point:

“In lines 114-115, the author says "the ridge waveguide should be restricted to single-mode propagation, i.e., h/H and W/H should not be too small." Is this not a sufficient explanation? Normally, to suppress higher-order modes, the core size should be reduced. However, too small a core size may cause cutoff.”

Response:

Thanks for this comment. According to the analysis in the second paragraph in Part 2, in order to distinct the energy distribution of two fundamental polarized modes in the ridge waveguide, h/H and W/H should be set as small as possible. Reducing h/H means increasing the waveguide height, which can allow higher-order modes propagation. Therefore, in order to guarantee single- mode propagation, the value of h should have a lower limit, which is around 160nm for the 600nm-thick r-TiO2 thin film. Although the waveguide dimension should be reduced to suppress higher-order modes, too small core size would affect the single-mode propagation in the waveguide. Therefore, the value of W should also have a lower limit, which is about 250nm. With these threshold values of core size, single-mode propagation can be allowed in the fabricated r-TiO2 waveguide.

According to this comment, the sentences “In addition, small ridge waveguide structures can increase the difficulty of both fabrication and testing of the optical waveguide. Thus, the r-TiO2 ridge waveguide structural parameters should be appropriately tailored.” in the second paragraph of Part 2 has been replaced by “Too small h/H would lead to the waveguide height increasing, so higher-order modes interaction can occur. Meanwhile, too small W/H would make it difficult for light to propagate in the waveguide. Therefore, there should be a threshold value of both h/H and W/H to guarantee single-mode propagation in the r-TiO2 ridge waveguide, and the r-TiO2 ridge waveguide structural parameters should be appropriately tailored.” We also add “low loss” in line 127 and delete “and W/H” in line 122. These modifications are marked in red.

Once again, thank you very much for your comments and suggestions.

Yours sincerely,

Yujie Ma

Reviewer 2 Report

I have the following suggestions:

1) In the title full form of  "r" i.e., "rutile" should be mentioned. 

2) As this is a numerical-based paper, a detailed explanation of the calculation of bending losses as shown in Figure 1, should be mentioned which is used in the simulation tool. Explain the model. Other conditions such as meshing, excitation port, boundary conditions should be added. 

3) What is the radius of the bend used in Figure 1? Is it a strip waveguide or a rib? Is it a 2D model or 3D? Explain in detail. 

4) The schematic of the waveguide should be separately added in Figure 1. 

5) What is the significance of PBS at the visible wavelength range? Where it can be used? Because in telecommunication applications (PDM), infrared wavelengths are used such as 1310 nm and 1550 nm.

6) In Figure 4, the PBS is proposed in the form of a coupler, then why in Figure 6, MRR-based PBS is proposed? What is the reason for proposing two configurations? 

7) I suggest the author provide a proposed fabrication method for the device. 

8) A comparison table should be provided at the end of the paper to compare the device performance of the proposed structure with already existing PBS based on different platforms and mechanisms, such as 10.1088/1555-6611/aadf18;  https://doi.org/10.1038/srep02206; https://doi.org/10.1364/OE.405375

ok

Author Response

Dear Reviewer:

Thank you very much for your letter referring to the manuscript of "Polarization splitting at visible wavelengths with r-TiO2 ridge waveguide" (ID: 2436654). We have checked and modified our manuscript according to the comments from reviewers. The detail of modification is listed as follows:

For the comments proposed by reviewer 2:

  1. The first review point:

“In the title full form of  "r" i.e., "rutile" should be mentioned.”

Response:

Thanks for the review’s careful review. The “r” has been changed to “rutile” in the title, marked in red.

  1. The second review point:

“As this is a numerical-based paper, a detailed explanation of the calculation of bending losses as shown in Figure 1, should be mentioned which is used in the simulation tool. Explain the model. Other conditions such as meshing, excitation port, boundary conditions should be added. ”

Response:

Thanks for the reviewer’s comments. A detailed description of the simulation conditions is necessary. We add “To calculated the bending loss of polarized light in the curved r-TiO2 waveguide, the model conducted by FDTD solutions from Lumerical software and 3D-FDTD algorithm were used for simulation, and the detailed calculation mechanism is shown in Ref. [52,53]. The light source in Fig. 1(c) and (d) is the fundamental TE0 mode and fundamental TM0 mode, respectively. The mesh accuracy is 20nm, and all boundary conditions are set to be PML conditions.” at the lines from 135 to 140 in the third paragraph of Part 2, marked in red.

  1. The third review point:

“What is the radius of the bend used in Figure 1? Is it a strip waveguide or a rib? Is it a 2D model or 3D? Explain in detail.” 

Response:

Thanks for the reviewer’s careful review. The radius of the bending waveguide in Figure 1 is 4um. The waveguides in Figure 1(c) and (d) are both in rib structure, and the cross-section of the curved waveguide is shown in Figure 1(a). We use a 3D model to simulate the optical transmission properties of polarized light in the curved waveguide. To supplement the above content, we replace the sentence “Energy distribution of the fundamental modes propagating in a bent waveguide for the TE0 (c) and TM0 (d) modes.” with “Energy distribution of the fundamental polarized modes propagating in a bent rib waveguide with the cross section shown in (a) for TE0 (c) and TM0 (d) modes, respectively. The radius of the bent waveguide is 4μm.”, in the legend of figure 1, and add the sentence “According to the simulation results of the optical transmission in the r-TiO2 ridge waveguide based on the 3D-FDTD model,” in the first paragraph of Part 2, marked in red.

  1. The fourth review point:

“The schematic of the waveguide should be separately added in Figure 1. ”

Response:

Thanks for the reviewer’s advice. The corresponding modifications have been added in Figure 1. The modified image is as follows:

  1. The fifth review point:

“What is the significance of PBS at the visible wavelength range? Where it can be used? Because in telecommunication applications (PDM), infrared wavelengths are used such as 1310 nm and 1550 nm.”

Response:

Thanks for this comment. Integrated photonic platforms operating at visible wavelengths are of great interests for applications ranging from quantum optics and metrology to bio-sensing and biomedicine. For example, alkali and alkaline earth metals such as rubidium, cesium, calcium, and sodium, the key elements for modern precision optical frequency metrology, magnetometry, and quantum computation, have their atomic transitions in the visible and near-visible spectrum range. Furthermore, visible wavelength light is used for quantum state preparation, manipulation, and readout of color centers, quantum dots, and various quantum emitters in 2D materials. These applications will inevitably involve polarization sensitive devices, so it is necessary to develop PBS for visible wavelengths. Moreover, visible light polarization control on chip can applied in quantum systems and underwater datacom for advanced cooling schemes for quantum computing.

According to this suggestion, we add this section “Integrated photonic platforms operating at visible wavelengths are of great interest for applications ranging from quantum optics and metrology to bio-sensing and biomedicine [15-17]. Furthermore, visible wavelength light is used for quantum state preparation [18], manipulation [19], and readout of color centers [20], quantum dots [21], and various quantum emitters in 2D materials [22,23]. These applications will inevitably involve polarization sensitive devices, so it is necessary to develop on-chip PBSs for visible wavelengths. For example, visible light polarization control on chip can be applied in quantum systems and underwater datacom for advanced cooling schemes for quantum computing [24].”  at the end of the first paragraph in the Introduction part to emphasize the application significance of PBS in visible wavelengths. The added content is marked in red.

  1. The sixth review point:

“In Figure 4, the PBS is proposed in the form of a coupler, then why in Figure 6, MRR-based PBS is proposed? What is the reason for proposing two configurations?” 

Response:

Thanks for the reviewer’s comment. In this paper, we propose a novel polarization separation mechanism based on ridge waveguide. In order to verify the feasibility of this polarization splitting mechanism, two kinds of on-chip PBSs were designed and their optical transmission properties were numerically analyzed. Based on these simulation results in this paper, we will conduct experiments on the fabricated r-TiO2 thin films to realize the proposed PBS on DC structure and the MRR-based PBS, and test their polarization separation properties in our future research work. Therefore, the proposed on-chip PBSs based on both DC structure and MRR structure can indicate that the polarization separation mechanism can be used to design multi-structure on-chip polarization beam splitting devices and clarify the principle of the polarization separation mechanism fully.

  1. The seventh review point:

“I suggest the author provide a proposed fabrication method for the device.” 

Response:

Thanks for the review’s suggestion, and we add the section “Electron beam lithography (EBL) and inductively coupled plasma etching (ICP) are going to be used to fabricate the ridge waveguide structure on the r-TiO2 thin film, and the fabrication process is briefly demonstrated here. Firstly, chromium and photoresist thin films are deposited on the sample surface successively, and the waveguide pattern is exposed and developed on the photoresist. Then the pattern of the photoresist will be transferred to the r-TiO2 thin film using ICP etching technology, and finally the etching effect will be observed using scanning electron microscope (SEM)” at the lines from 151 to 127 in the third paragraph of Part 2, marked in red.  

  1. The eighth review point:

“A comparison table should be provided at the end of the paper to compare the device performance of the proposed structure with already existing PBS based on different platforms and mechanisms, such as 10.1088/1555-6611/aadf18; https//doi.org/10.1038/srep02206;  https//doi.org/10.1364/OE.405375”

Response:

Thanks for the reviewer’s suggestion, and we add a comparison table at the end of this paper to illustrate the on-chip PBS device performance based on different structures and mechanisms, as shown below:

A brief summary of previously reported PBSs based on different structures is presented in the below Table l. It can be seen that the proposed PBSs in this work is working at visible wavelengths, which is different from previous PBSs.

Table 1. Comparison of the PBSs based on different structures.

Sturctures

Length

(μm)

ER(dB)

EL(dB)

Bandwidth(nm)

 LSPs [58]

1.1

20.69 (TE)

20.33 (TM)

0.249 (TE)

0.671 (TM)

(1450nm-1650nm)200nm

(ER>20dB,TE)

(ER>12dB,TM)

  DCs [59]

11

20.32 (TE)

15.4 (TM)

1.69 (TE)

2.72 (TM)

(1530nm-1565nm)35nm

(ER>10.6dB,TE)

(ER>14.6dB,TM)

 ADCs [60]

16

26.7 (TE)

21.3 (TM)

0.5 (TE)

0.2 (TM)

(1480nm-1620nm)140nm

(ER>10dB,TE)

(ER>15dB,TM)

SWGs [61]

160

25 (TE)

31 (TM)

<1

(1450nm-1634nm)185nm

(ER>20dB,TE)

(1490nm-1575nm)85nm

(ER>20dB,TM)

ADCs [62]

31

55 (TE)

55 (TM)

0.066 (TE)

0.2 (TM)

(1500nm-1600nm)100nm

(ER>29.4dB,TE)

(1527nm-1587nm)60nm

(ER>20dB,TM)

DCs (this work)

90

38 (TE)

43 (TM)

2 (TE)

0.45 (TM)

(610nm-660nm)50nm

(ER>40dB,TE)

(ER>22dB,TM)

MRRs(this work)

48

45 (TE)

58 (TM)

2.5 (TE)

0.7 (TM)

(610nm-660nm)50nm

(ER>40dB,TE)

(ER>40dB,TM)

Once again, thank you very much for your comments and suggestions.

Yours sincerely,

Yujie Ma

Round 2

Reviewer 2 Report

I am willing to accept the paper in its current form. 

ok